# Mechanisms of Cannabis Growth Promotion by *Bacillus velezensis* S141

**DOI:** 10.3390/plants13212971

**Published:** 2024-10-24

**Authors:** Phirom Aunkam, Surachat Sibponkrung, Sirawich Limkul, Tuangrak Seabkongseng, Kanjana Mahanil, Kamolchanok Umnajkitikorn, Nantakorn Boonkerd, Neung Teaumroong, Shusei Sato, Panlada Tittabutr, Pakpoom Boonchuen

**Affiliations:** 1School of Biotechnology, Institute of Agricultural Technology, Suranaree University of Technology, Nakhon Ratchasima 30000, Thailand; 2Center of Excellent in Agricultural Product Innovation, Suranaree University of Technology, Nakhon Ratchasima 30000, Thailand; 3Institute of Research and Development, Suranaree University of Technology, Nakhon Ratchasima 30000, Thailand; 4School of Crop Production Technology, Institute of Agricultural Technology, Suranaree University of Technology, Nakhon Ratchasima 30000, Thailand; 5Graduate School of Life Sciences, Tohoku University, Sendai 980-8577, Japan

**Keywords:** *Cannabis sativa* L., *Bacillus velezensis* S141, plant-growth-promoting bacterium (PGPB)

## Abstract

*Cannabis sativa* L. has a variety of uses, including fiber production, food, oil, and medicine. In response to environmental concerns regarding chemical fertilizers, *Bacillus velezensis* S141 was examined as a plant-growth-promoting bacterium (PGPB) for cannabis. This study evaluated the effects of S141 on cannabis growth and utilized transcriptomic analysis to identify the responsive pathways. Inoculation with S141 significantly increased growth in laboratory and field environments, with most of the bacteria residing in the leaves, followed by the stems and roots, as determined by quantitative polymerase chain reaction (qPCR). Transcriptomic analysis revealed 976 differentially expressed genes. Upregulated genes were associated with metabolism, cellular processes, and catalytic activities, especially in the biosynthesis of phenylpropanoid, plant–pathogen interactions, and hormone signaling pathways. S141 mutants deficient in the production of auxin and cytokinin displayed reduced growth enhancement, which affirmed the roles of these hormones in cannabis development. These findings emphasize the potential of S141 as a sustainable growth promoter for cannabis and provide insights into the underlying pathways it influences.

## 1. Introduction

*Cannabis sativa* L., more commonly known as marijuana, belongs to the Cannabaceae family and is widely cultivated around the globe [1]. Cannabis has served diverse purposes since ancient times; it has acted as a folk medicine, a psychoactive drug, and a material in the production of textiles and rope [2,3,4]. With the global interest in cannabis cultivation on the rise, optimizing plant health, productivity, and cultivation methods has become critically important. Among several options, organic cannabis farming has gained popularity due to concerns connected to the health, sustainability, and quality of the produce [5]. One promising approach for enhancing the growth and overall health of cannabis plants involves the use of plant-growth-promoting bacteria (PGPBs).

In contemporary agriculture, addressing the dual challenges of increasing food demand and environmental sustainability has become imperative. PGPBs have emerged as sustainable bioresources capable of enhancing crop productivity and resilience in various agroecosystems. These bacteria not only improve nutrient availability and uptake efficiency but also enhance root development and stimulate systemic resistance in plants, thereby mitigating biotic and abiotic stresses [6,7]. Moreover, some PGPBs can produce enzymes like 1-aminocyclopropane-1-carboxylic acid (ACC) deaminase, which is able to alleviate plant stress by reducing ethylene stress levels [8]. In the context of cannabis, the plant faces numerous challenges, such as nutrient depletion and susceptibility to pathogens. Incorporating PGPBs into cannabis farming has shown promise for enhancing plant growth, yield, and resilience. For instance, the supplementation of *Pseudomonas* and *Bacillus* during cannabis cultivation was found to improve plant health and productivity by suppressing soil-borne pathogens and optimizing nutrient availability [9]. The efficacy of *Azospirillum brasilense B.* in enhancing root development and nutrient uptake in cannabis has been noted, while *Bacillus* species have demonstrated benefits through enhanced nitrogen availability and phytohormone production [10]. Hence, integrating PGPBs into cannabis cultivation could support sustainable agriculture and enhance the crop’s economic and environmental sustainability.

*Bacillus velezensis* [11] has emerged as a critical species of PGPB with significant uses in agriculture. Thanks to its ability to produce a range of bioactive compounds such as indole-3-acetic acid (IAA), siderophores, and antimicrobial peptides, *B. velezensis* plays essential roles in enhancing plant growth, nutrient uptake efficiency, and stress tolerance in various crops [12,13]. This species has been thoroughly studied for its biocontrol features against phytopathogens, including fungi and bacteria, thus reducing dependency on chemical pesticides and promoting sustainable agricultural practices [12,13]. Furthermore, *B. velezensis* has shown effectiveness in improving soil health, decontaminating soils, and enhancing crop yields in diverse agroecosystems [14,15]. As researchers continue to decipher its mechanisms of action and optimize its applications, *B. velezensis* is poised as a multifaceted tool for modern agriculture, addressing issues of food security and environmental sustainability.

One of the *B. velezensis* strains, specifically S141, has gained attention for its versatile applications in agriculture. It was isolated from the soybean (*Glycine max* (L.) Merr. (Fabaceae)) rhizosphere, which functions as a robust plant-growth-promoting rhizobacterium and a biocontrol agent against phytopathogens [16]. Studies have reported its proficiency in enhancing crop yields and disease resistance in several crops, including rice, soybean, and maize [16,17,18]. However, research regarding the functional properties and utilization of *B. velezensis* S141 in cannabis, especially concerning growth-developing impacts, is scarce.

In this study, we investigated the use of S141 as a PGPB in cannabis cultivation and the potential mechanisms that activate the biological pathways in cannabis. This was achieved through a transcriptomic analysis of the bacterium’s influence.

## 2. Results

### 2.1. The Colonization of Bacillus velezensis S141 After Cannabis Inoculation

Prior to inoculating the cannabis with *B. velezensis* S141, cannabis seedlings were grown for 10 days in vitro. At 28 days after inoculation, the plant tissues, including the leaves, stems, and roots, were harvested, surface-sterilized, and subjected to genomic DNA extraction. These extracts were then used as templates for qPCR analysis to confirm the presence of S141. The highest number of S141 copies were found in the leaves (4.89 × 10^5^ copies/20 ng DNA), followed by the stems (3.82 × 10^4^) and roots (1.17 × 10^4^) in the S141-inoculated cannabis. No S141 was found in the non-inoculated group (Figure 1). These findings suggest that S141 is an endophyte and PGPB in cannabis.

### 2.2. The Improved Growth Performance of the Bacillus velezensis S141-Inoculated Cannabis: Laboratorial Cultivation

To explore the influence of S141 on cannabis growth at the laboratory scale, this study varied the amounts of S141 added to the cannabis plants in Leonard’s jars. We then monitored the quantity of chlorophyll and the health index (HI). To initially evaluate the growth influenced by S141 supplementation, images of the plants were taken at different points after S141 inoculation. From 3 to 14 day after inoculation (DAI), the overview of plant characteristics displayed noticeable increases in root length, plant height, root dry weight, and total dry weight when compared to the control (Figure 2A–F). Similarly, S141 supplementation led to a significant increase in chlorophyll content, as measured by SPAD units. At 28 DAI, all experimental groups inoculated with S141 showed an approximately 20% higher chlorophyll content compared with the control (Figure 2C). Moreover, the HI values of the 10^4^, 10^6^, and 10^8^ groups at 3, 5, 7, and 14 DAI were about 2-fold higher than that of non-inoculated ones, while no significant differences were observed at 21 and 28 DAI (Figure 2D). For root dry weight and total dry weight, only groups supplemented with S141 at 10^6^ and 10^8^ CFU/mL displayed approximately 2-fold-higher mg/plant in both categories (Figure 2E,F).

### 2.3. The Improved Growth Performance of the Bacillus velezensis S141-Inoculated Cannabis: Greenhouse Cultivation

To evaluate the impact of S141 on the growth performance of cannabis in a greenhouse, an experiment was designed using the same protocols as stated above. The plants were transferred into pots containing different types of soils and various levels of fertilizer and nurtured for 65 days after S141 inoculation. Upon examining the images collected at the end of the experiment, it was revealed that the growth of cannabis treated with S141 exhibited notable differences in the 10^4^ to 10^8^ CFU/mL group cultivated in both boiled soil supplemented with normal and low fertilizer (Figure 3A–H). As for the dry weight of the leaves, roots, stems, and branches, as well as the total plant, no significant different was observed between the control and S141 inoculation at 10^4^ CFU/mL. For S141 inoculation at 10^6^ and 10^8^ CFU/mL, we found an approximately 2- to 4-fold increase in weight compared to the non-inoculated group (Figure 3E–H). This trend is similar to that observed with laboratory cultivation.

### 2.4. Transcriptomic Analysis, GO Terms, and KEGG Pathways

To uncover the molecular mechanism by which the S141 inoculum promoted cannabis growth, a transcriptomic analysis was conducted from three biological replications of each group. About 40 million total reads were generated from six libraries. Of these, clean reads were identified from Q20, constituting more than 96% (Appendix A). A total of 976 DEGs were detected, the data for which were filtered at a false discovery rate (FDR) of less than 0.05 and an absolute Log_2_ (fold change) value of greater than one for comparison between the non-inoculated and inoculated groups. The findings indicated that the number of up-regulated genes exceeded that of the downregulated ones, 606 to 370, respectively (Figure 4A). The DEGs associated with molecular functions, cellular components, and biological processes were then examined in GO categories. Increased activities in metabolic processes, cellular processes for biological processes, cellular anatomical entities for cellular components, and catalytic activity and binding for molecular functions were found (Figure 4B). Finally, KEGG analysis led to the identification of potential growth-promoting pathways in cannabis impacted by S141, such as the biosynthesis of phenylpropanoid, plant–pathogen interaction, and plant hormone signal transduction (Figure 4C).

### 2.5. Gene Expression Validation by qRT-PCR

A total of 18 randomly chosen genes related to cannabis growth were studied using qRT-PCR. These included auxin-responsive protein SAUR50 (SAUR50), xyloglucan endotransglucosylase/hydrolase protein 25 (XEHP25), germin-like protein 2-1 (GLP2-1), ABC transporter G family member 29 (ABC29), carboxylesterase 120 (CBL120), receptor-like protein EIX1 (EIX1), ethylene-responsive transcription factor CRF5 (CRF5), indole-3-acetic acid (IAA2), response regulator (ARR5), response regulator (ARR12), caffeic acid 3-O-methyltransferase (CAO), UDP-glucose flavonoid 3-O-glucosyltransferase 7-like (UDP), ethylene-response factor C3 (ERFC3), tetrahydrocannabinolic acid synthase (THCAS), cannabidiolic acid synthase (CBDAS), G-type lectin S-receptor-like serine/threonine-protein kinase At2g19130 (GTLS), leucine-rich repeat receptor-like protein kinase At5g49770 (LRLK4), and pathogenesis-related protein 1A-like (PRP-1A). The latter was used to validate the RNA-Seq results (Figure 5).

Significantly higher expressions of *THC*, *GLP2-1*, *XEHP25*, *IAA2*, and *SAUR50* (about 2- to 6-fold) was found in all tissues—roots, stems, and leaves—than in the uninoculated group. The *UDP*, *CBD*, *CBL120*, *ARR12*, *ARR5*, and *CRF5* were significantly upregulated 2- to 10-fold in stems and leaves when compared to the control. An increase of about 8-fold in *ERFC3* and *EX11* expressions was observed only in leaf tissue. Root and stem tissues showed an about 2.5-fold upregulation of *ABC29* expression. *GTLS*, *LRLK4*, and *PRP-1A* were significantly downregulated around 2- to 5-fold. It is worth noting that the expression pattern determined from the RNA sequencing (RNA-Seq) data was similar to the expression pattern of the selected DEGs determined from the RT-qPCR results.

### 2.6. The Effects of Bacillus velezensis S141 Mutants on the Foi Thong Suranaree 1 Cannabis Strain Growth Under Laboratory Conditions

S141 was previously characterized by its genetic repertoire, which comprises various genes associated with plant growth promotion and biocontrol activities. The beneficial effects can be traced to key genes, including those that encode enzymes for indole-3-acetic acid (IAA) synthesis, such as *yhcx*, *IPyAD*, and *dhaS*. These genes play an important role in IAA production from indole-3-pyruvic acid. In addition, the cytokinin biosynthesis pathway includes *IPT* and *IPI* genes. This pathway responds to the *IPI* gene encoding the isopentenyl pyrophosphate isomerase (IPI) enzyme, which converts isopentenyl pyrophosphate (IPP) into dimethylallyl pyrophosphate (DMAPP). This DMAPP then acts as a substrate for the enzyme isopentenyl transferase (IPT), which is responsible for cytokinin biosynthesis [15].

To examine the impact of the genes related to the production of plant hormones in S141, *B. velezensis* S141 mutants (*IpyAD*, *dhas*, *yhcx*, *IPT*, and *IPI*) and wild-type S141 were inoculated onto cannabis plants at a concentration of 10^6^ CFU/mL. At 7–14 DAI, no significant differences in chlorophyll content were observed across all experimental groups inoculated with S141 and S141 mutants when contrasted with the control group (uninoculated). Investigations into the characteristics of the plants from S141 and S141 mutants (*dhas*, *yhcx*, and *IPI*) revealed noticeable increases in root length, plant height, root dry weight, total dry weight, and HI compared to those of the uninoculated group (Figure 6A–F).

Interestingly, in groups inoculated with S141 mutant genes (*IpyAD* and *IPT*), no significant changes were observed in the root length, plant height, root dry weight, total dry weight, or HI when compared to the uninoculated group. These findings suggest that the genes encoding for plant hormones in S141 (*IpyAD* and *IPT*) play a significant role in promoting growth in cannabis.

## 3. Discussion

*Bacillus velezensis* is recognized for its diverse applications as a PGPB and a plant-growth-promoting rhizobacterium (PGPR) [21,22,23]. *B. velezensis* has been identified as a PGPB in cannabis, similar to in this study, which discovered the highest S141 copy number in surface-sterilized tissues and leaves, followed by the stems and roots of inoculated cannabis. No S141 was present in the non-inoculated group. This result highlights that S141 might be an endophytic bacterium in cannabis (Figure 1). Several strains of *B. velezensis* are known for their capacity to enhance plant growth through various mechanisms. *B. velezensis* FZB42, renowned for producing phytohormones such as indole-3-acetic acid (IAA), improves nutrient uptake, leading to increased biomass and stress tolerance in crops like *Arabidopsis thaliana* (L.) Heynh. (Brassicaceae) [12,24]. *B. velezensis* 83 promotes maize growth by stimulating root development through phosphate solubilization, auxin production, and volatile organic compound emissions, thus boosting overall plant vigor [21]. Moreover, *B. velezensis* BS1 has demonstrated biocontrol efficacy against fungal pathogens in pepper plants, contributing to enhanced plant health and growth by reducing disease stress [25]. S141 is seen as a promising plant-growth-promoting rhizobacterium with multifaceted benefits for agricultural systems. It enhances plant growth, nodulation, and nitrogen fixation efficiency, particularly in collaboration with crops like soybean, highlighting its potential as a beneficial tool in sustainable agriculture [16]. Additionally, S141 showcases the capacity to boost arbuscular mycorrhizal symbiosis, increasing nutrient absorption and usage in host plants. By activating key plant marker genes linked to mycorrhizal symbiosis and upregulating the genes essential for nutrient absorption, S141 contributes to improved plant performance and resilience. Its effect on the cost–benefit balance in mycorrhizal symbiosis underscores its complex interactions within plant–microbe systems [26]. In this study, S141 inoculation substantially enhanced cannabis growth, particularly at an optimal concentration of 10^6^ CFU/mL, as demonstrated by increased trunk circumference, height, chlorophyll content, and dry weight (Figure 2). Furthermore, greenhouse experiments with four separate soil conditions, including soils treated with boiled water and untreated soils, combined with either normal or low fertilizer levels, validated these findings. The results showed the practical benefits, such as improved growth parameters and reduced fertilizer needs, of treatments with this bacterium (Figure 3). These findings emphasize the potential of *B. velezensis* S141 to advance cannabis cultivation practices, promote cannabis growth, encouraging further exploration of its action mechanisms and optimization for extensive agricultural applications.

Transcriptomes have been utilized in cannabis to decipher how the plant responds and defends itself against biotic and abiotic stresses [27,28,29]. This study found transcriptomic analysis useful in identifying the plant genes associated with S141 inoculation (Figure 4). The GO term analysis showed that the upregulation of biological processes such as metabolic process, cellular process, localization, and biological regulation was higher than downregulation following *B. velezensis* inoculation. This suggests a connection to the network of metabolic pathways responsible for growth, development, and environmental responses [30]. Therefore, it is plausible that the differential gene expressions in S141-inoculated *C. sativa* might relate to the growth rate of *C. sativa*. Consequently, the KEGG pathways describe metabolism-associated biological processes in the biosynthesis of phenylpropanoid, plant–pathogen interaction, and plant hormone signal transduction. As previously reported, MdMYB88 and MdMYB124 directly regulate the accumulation of the metabolites involved in the phenylpropanoid pathway by modulating the expression of MdCM2, which aids in defending against pathogens and tolerating drought stress [31]. Effectors in plant–pathogen interactions operate in key ways: they break through physical barriers to invade the plant, create an environment that supports their survival inside the plant, and even employ tactics to evade or deceive plant defenses, while weakening the plant’s immune responses [32]. Our study observed such behaviors in plant–pathogen interactions after inoculation with the bacterium, which suggests that these interactions might develop and respond to the environment of the organismic system of plants. In the realm of plant hormone signal transduction pathways, membrane or transmembrane receptors display specificities and diversities. Acting as gateways, these receptors play an essential role in recognizing various hormones and transmitting signals into the cell [33]. Recent studies showed that the upregulation of phytohormone signal transduction is crucial in the response of Jerusalem artichoke (*Helianthus tuberosus* L., (Asteraceae)) seedlings to salt stress. The genes studied included those for abscisic acid, auxin, ethylene, and jasmonic acid [34]. Plant growth regulators have been studied for their ability to enhance the growth and yield of rice under high-temperature conditions both day and night, with the most significant production found via hormone treatments that boosted photosynthesis [35]. Our results suggested that plant hormone signal transduction was represented by KEGG pathways during *B. velezensis* inoculation in *C. sativa*. This implies that biological processes related to gene metabolism, previously identified as pivotal in growth pathways, could be highly beneficial for plant growth in conjunction with plant–pathogen interactions and plant hormone processes.

In this study, we used qRT-PCR to explore the expression profiles of 18 genes related to cannabis growth (Figure 5), with a particular focus on their modulation by phytohormones involved in developmental processes. Genes such as *SAUR50*, *IAA2*, and *ARR5* showed significant upregulation across all tissues (root, stem, and leaf). Previous studies have shown that auxins like *IAA2* are essential in root and shoot development by regulating cell elongation and differentiation [36]. SAUR proteins, such as *SAUR50*, have been associated with auxin-mediated growth responses, suggesting their conserved role across plant species [37]. The upregulation of these genes in our study is consistent with their known functions in promoting vegetative growth, which is vital for cannabis cultivation and biomass production. Additionally, the differential expression of genes like *UDP*, *CBD*, *CBL120*, *ARR12*, *ARR5*, and *CRF5* predominantly in the stems and leaves underscores the tissue-specific regulation of growth-related pathways. For example, cytokinin-responsive genes (such as *ARR5* and *ARR12*) are known to affect shoot branching and leaf senescence [38], potentially impacting cannabis plant architecture and yield. The upregulation of CBD-related genes in the stems aligns with the biosynthesis of cannabinoids, essential compounds with pharmaceutical relevance, thus emphasizing the economic significance of understanding their regulatory mechanisms in different plant tissues [2]. Comparative analyses with previous studies emphasize both the evolutionarily conserved and divergent regulatory mechanisms of phytohormone-responsive genes across plant species. Studies on germin-like proteins (e.g., *GLP2-1*) involved in cell wall modification and growth regulation further highlight their conserved roles in hormonal signaling pathways affecting plant architecture and development [39,40]. The observed downregulation of genes like *GTLS* and *LRLK4* suggests their potential roles in modulating receptor-mediated signaling pathways and stress responses, indicative of complex hormonal crosstalk governing growth and development in cannabis and related species [41,42]. These findings underscore the intricate interplay between hormonal signals and gene regulation, emphasizing the importance of understanding these mechanisms for optimizing crop productivity and metabolic pathways in cannabis cultivation. Future research could examine in greater detail the specific molecular interactions and signaling pathways involved in phytohormone-mediated growth processes, thereby facilitating targeted breeding strategies and biotechnological advancements in cannabis agriculture.

The roles of the genes involved in the production of plant hormones, auxin and cytokinin, in bacteria are pivotal for understanding and optimizing plant–microbe interactions. Genes such as *yhcx*, *IPyAD*, and *dhaS*, which are integral to auxin biosynthesis, enhance plant growth by promoting root elongation, nutrient uptake, and stress resistance [43,44]. Likewise, cytokinin biosynthesis genes like IPT and IPI contribute to shoot growth, cell division, and delay in leaf senescence [45,46]. Mutations in these genes can drastically reduce hormone production, leading to reduced bacterial colonization, plant growth promotion, and disease resistance. In prior studies, co-inoculation of *Bradyrhizobium diazoefficiens* F (Xanthobacteraceae) strain USDA110 with S141Δ*yhcx* and S141Δ*IPI* reduced the number of nodules, indicating the significant impact of *yhcx* and *IPI* on promoting soybean growth [16]. In this study, we elucidated the distinct roles of the genes involved in plant hormone production in cannabis growth via the investigation of S141 and its mutant strains. Significant differences in growth parameters emerged between the inoculated and control groups. Specifically, both wild-type S141 and mutants possessing *dhaS*, *yhcx*, and *IPI* genes exhibited marked increases in root length, plant height, root dry weight, total dry weight, and the health index (HI), underlining the importance of these genes in promoting robust cannabis growth (Figure 6A–F). Conversely, mutants lacking *IpyAD* and *IPT* genes displayed no significant improvements in these metrics compared to the controls, implying a crucial role of *IpyAD* and *IPT* in mediating cannabis growth promotion. These findings contribute to our understanding of how specific genetic elements in S141 enhance plant growth, potentially through the modulation of the hormone signaling pathways essential for developmental processes in cannabis.

This study offers substantial evidence of the growth-promoting effects of S141. However, further research is necessary to thoroughly understand its mechanisms of action and to optimize its use in cannabis cultivation. Future research could concentrate on understanding the distinct metabolic pathways and signaling mechanisms engaged in plant–microbe interactions, as well as investigate potential synergies with other beneficial microorganisms or agricultural inputs. Moreover, conducting field trials over several growing seasons and under varied environmental conditions could help confirm the effectiveness and consistency of S141 inoculation across different cannabis cultivars and cultivation practices (Figure 7).

## 4. Materials and Methods

### 4.1. The Statement of Ethics

According to the risk levels prescribed for pathogens and animal toxins in “The Risk Group of Pathogen and Animal Toxin (2017)” by the Department of Medical Sciences, Ministry of Public Health, Pathogen and Animal Toxin Act (2015), and Biosafety Guidelines for Modern Biotechnology, BIOTEC (2016), the biosecurity aspects of this study were reviewed and approved by Suranaree University of Technology (approval number: SUT-IBC-003/2023). Furthermore, authorization for cannabis cultivation and production was acquired (license: 13/2563), and the processing of plant materials was carried out under the supervision of the Thailand Food and Drug Administration (TFDA). The procedures followed the standard operating procedures (SOPs) for legally compliant cannabis production.

### 4.2. Bacterial Strains and Growth Conditions

*Bacillus velezensis* S141 was cultured in nutrient broth at 30 °C for 18 h, while the mutants (*IpyAD*, *dhas*, *yhcx*, *IPT*, and *IPI*) were grown in Luria–Bertani medium under the same conditions, achieving cell densities between 10^4^ and 10^8^ CFU mL^−1^. The media were supplemented with erythromycin (1 μg mL^−1^), kanamycin (10 μg mL^−1^), phleomycin (8 μg mL^−1^), and spectinomycin (100 μg mL^−1^) to prepare inoculum containing bacterial concentrations of 10^4^, 10^6^, and 10^8^ CFU mL^−1^. After incubation, cells were collected by centrifuging at 4000× *g* for 10 min, washed with sterile 0.85% (*w*/*v*) NaCl to remove residual media, and then resuspended in sterilized deionized water to achieve the target concentrations (10^4^, 10^6^, and 10^8^ CFU/mL). Appendix A provides a list of the bacterial strains used in this study.

### 4.3. Cannabis Sativa Strain Used in This Study

Foi Thong Suranaree 1 is a cannabis strain that was bred and developed by the Hemp-Cannabis Research, Production, and Utilization Project’s “Sub-Project Breeding and Testing Strains”. This project is under the umbrella of the Center of Specialization in Agricultural Product Innovation at the School of Agricultural Technology, Suranaree University of Technology. The strain demonstrates robust growth, rapid development, and sizable leaves and stems. Additionally, it offers effective canopy division and a strong resistance to diseases and pests. This strain is also notable for its prompt flowering and impressive adaptability to various environmental conditions. With high yield potential, Foi Thong Suranaree 1 provides a compelling opportunity for farmers interested in commercial cultivation.

### 4.4. Investigation of Plant Growth Promotion

Cannabis was grown under laboratory and greenhouse conditions. In the laboratory setting, 200 cannabis seeds were surface-sterilized using 70% (*v*/*v*) ethanol, 3% (*v*/*v*) sodium hypochlorite, and sterile water. A seedling (1 per pot) was subsequently transplanted into sterilized vermiculite within Leonard’s jars that contained AB fertilizer solution (EC: 1.8–2 mS/cm, pH 6.5–7). This cultivation took place under controlled sterile conditions, maintaining a temperature of 25 ± 2 °C, 12-h light/12 h dark photoperiods, and 50% humidity. Inoculations of S141 at various concentrations, 10^4^, 10^6^, and 10^8^ CFU mL^−1^, were conducted after 10 days after seedling transplantation, establishing four groups of five seedlings each: a control group that received sterile water and treatment groups that received varying concentrations of the S141 inoculum. Plant growth was assessed by measuring chlorophyll content, health index, fresh weight, and dry weight at 3, 5, 7, 14, and 28 days after inoculation (DAI).

For the greenhouse cultivation study, four distinct soil conditions were prepared. These included soils treated with boiled water and untreated soils, each combined with either normal or low fertilizer levels, as outlined in Appendix A. Cannabis seeds were subjected to surface sterilization as described above before being planted in 1 seedling per container with these specific soil treatments and cultivated for 14 days. After this period, 1 mL of S141 inoculum (10^4^, 10^6^, and 10^8^ CFU) was applied to the root base of each cannabis plant monthly. Sixteen experimental groups of five each were established, mirroring as the laboratory setup: a control group and a treatment group, with the latter receiving different concentrations of the S141 inoculum. The plant growth was assessed by measuring their fresh and dry weights at 65 DAI.

### 4.5. Evaluating the Chlorophyll Content and Health Index (HI) of the Cannabis

The level of chlorophyll in the cannabis was monitored by a chlorophyll meter (SPAD-502Plu, Konica Minolta, Chiyoda, Japan), and the health index (HI), a tool for evaluating factors that directly impact the survival and growth of cannabis plants, was calculated using the equation mentioned below [47]. These parameters were recorded after the inoculation of wild-type S141 and S141 mutants (*IpyAD*, *dhas*, *yhcx*, *IPT*, and *IPI*). The measurements of chlorophyll content, stem diameter, stem height, and dry weight were documented at 3, 5, 7, 14, 21, and 28 DAI.
$$Health\;index=\frac{Stem\;diameter}{Stem\;height}\times Dry\;weight$$

### 4.6. RNA and DNA Extraction

After a 28-day cannabis cultivation period, plant tissues—including roots, stems, and leaves—were gathered and ground using a mortar and pestle under liquid nitrogen. Approximately 100 mg per sample was moved into a 1.5 mL microfuge tube; afterward, an RNeasy RNA Mini Kit (QIAGEN, Hilden, Germany) was utilized to extract and purify total RNA from these plant tissues. The quantity and quality of the RNA were evaluated using a Nanodrop 2000 Spectrophotometer (Thermo Scientific, Waltham, MA, USA) and agarose gel electrophoresis stained with RedSafe Nucleic Acid Staining Solution (iNtRON, Seongnam, Republic of Korea), respectively. For the extraction of bacterial DNA within the cannabis, plant tissues were steeped in 75% ethanol (*v*/*v*) for 1 min and subsequently rinsed with sterile distilled water. The samples were then exposed to a 3% sodium hypochlorite solution (*v*/*v*) for 2 min and cleaned again with sterile distilled water prior to DNA extraction. A QIAGEN PowerSoil DNA isolation kit was used to extract and purify the genomic DNA of the cannabis and bacteria. The quantity and purity of DNA were assessed as previously described.

### 4.7. Sequencing Analysis

The quality of RNA was determined using NanoDrop, Qubit 2.0, and Agilent 2100 systems before commencing the library construction. Six libraries were prepared from 2 μg each of the total RNA extracted from inoculated and non-inoculated cannabis tissues. These tissues had been exposed to S141 at a concentration of 10^6^ CFU/mL. BMKGENE, China, performed the sequencing. mRNA was isolated using Oligo(dT)-attached magnetic beads, and the enriched RNAs were randomly fragmented in a fragmentation buffer. First-strand cDNAs were synthesized from the fragmented RNAs using random hexamer primers, followed by second-strand synthesis with the addition of PCR buffer, dNTPs, RNase H, and DNA polymerase I. The cDNAs were then purified using AMPure XP beads. These underwent an end-repair procedure, with adenosine added to the end while ligated to adapters. The fragments were then selected using AMPure XP beads within the size range of 300–400 bp. The cDNA library was generated through several rounds of PCR on the cDNA fragments; the qualified library was sequenced using a high-throughput platform in PE150 mode. Clean data were obtained by filtering raw data to remove the adapter sequence and read of low quantity; these filtered data were then aligned with the Cannabis_sativa.cs10 reference genome (GCA_900626175.2). The differentially expressed genes (DEGs) were selected for further analysis based on *p*-values < 0.05 and absolute Log_2_ ratios ≥ 1. The DEGs were annotated in the differential expression analysis, and Gene Ontology (GO) and the Kyoto Encyclopedia of Genes and Genomes (KEGG) were then analyzed regarding the DEGs to assess the functional networks of gene products and genes within the pathways, respectively. We examined three types of GO annotation systems, which included biological process, molecular function, and cellular component. The annotation of DEGs in KEGG was built based on KEGG’s database of genes within pathways, including the metabolic pathways of carbohydrates, nucleotides, amino acids, and biological degradation of organics.

### 4.8. Gene Expression Analysis by Quantitative Real-Time PCR (qRT-PCR)

One microgram of total RNAs, extracted from the plant tissues of both non-inoculated and inoculated cannabis, was used as a template for cDNA synthesis using an iScript^TM^ cDNA Synthesis Kits (Bio-Rad, Hercules, CA, USA). Primers specific to selected candidate genes were designed using Primer3Plus software (https://www.primer3plus.com/index.html (accessed on 21 August 2024)) and are displayed in Appendix A. The qRT-PCR analysis was conducted using Luna Universal qPCR Master Mix (New England BioLabs, Frankfurt, Germany) and the CFX96 touch real-time PCR detection system (Bio-Rad). Each sample underwent triplicate examination, with amplification conditions as follows: 98 °C for 2 min and 40 cycles of 95 °C for 5 s and 60 °C for 30 s. Relative expression was calculated based on 2^−∆∆Ct^, with actin used as an internal control [19,20].

### 4.9. Estimation of the Number of Endophytic S141 in Different Plant Tissues

Quantitative real-time PCR (qRT-PCR) was used to determine the copy numbers of S141 quantitatively. The reaction was carried out using a CFX96 Touch™ Real-Time PCR Detection System (Bio-Rad) and Luna Universal qPCR Master Mix (New England Biolabs), in addition to specific primer pairs (Appendix A). The copy numbers were quantified according to a standard curve established using a plasmid containing the S141 sequence [48].

### 4.10. Investigation of the Cannabis Growth Profiles Upon Inoculation With Bacillus velezensis S141 Mutants

To evaluate the effects of plant hormones synthesized by S141 on cannabis growth profiles, we utilized mutants ∆*IpyAD*, ∆*dhas*, ∆*yhcx*, ∆*IPT*, and ∆*IPI*, as studied by Sibponkrung et al. [15]. Sterilized vermiculite, contained within Leonard jars, was used for cultivation, and all equipment was sterilized by autoclaving at 121 °C for 30 min preceding seedling transplant. Surface-sterilized cannabis seedlings were subsequently placed onto the sterilized vermiculite within culture trays. Ten days following transplantation, each seedling received 1 mL of S141 mutant strain (*IpyAD*, *dhas*, *yhcx*, *IPT*, and *IPI*) at a concentration of 10^6^ CFU/mL, while the control group received 1 mL of sterile water. Cultivation conditions were strictly controlled: temperature 25 ± 2 °C, photoperiod of 12 h light/12 h dark, and humidity at 50%. On days 7 and 14 after inoculation, fresh and dry weights of plants were recorded to assess growth patterns.

### 4.11. Statistical Analysis

A paired sample *t*-test was used to identify significant differences between the means of each group (*p* < 0.05). The data are presented as the average ± standard deviation (SD), which were derived from three biological replicates.

## 5. Conclusions

This study underscores the significant impact of *B. velezensis* S141 inoculation on the growth profiles and gene expression of *C. sativa* inoculated with S141. Both laboratory and greenhouse cultivation trials evidenced that S141 could positively impact various facets of cannabis growth, comprising increased stem size, height, chlorophyll content, and the dry weights of the leaves, stems, and roots. Furthermore, RNA sequencing analysis detected considerable modifications in gene expression, notably in metabolic processes, cellular components, and catalytic activities, underlining the intricate mechanisms governing the symbiotic relationship between S141 and cannabis. Moreover, pathway enrichment analysis signaled key pathways linked to plant growth and defense, accentuating the potential of S141 as a bioinoculant for enhancing cannabis cultivation practices.

## Figures and Tables

**Figure 1 plants-13-02971-f001:**
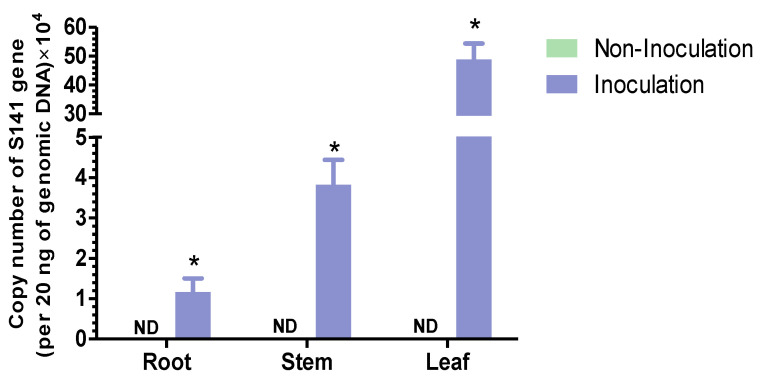
Localization of S141 in cannabis after inoculation with *B. velezensis* S141. Genomic DNA was collected from the leaf, stem, and root tissues at 7 days after inoculation. S141 copy levels were measured in triplicate using qRT-PCR. Data are presented as mean ± SD for n = 3 from three biological replications, with significant differences (*p* < 0.05) indicated by asterisks. ND denotes not detected.

**Figure 2 plants-13-02971-f002:**
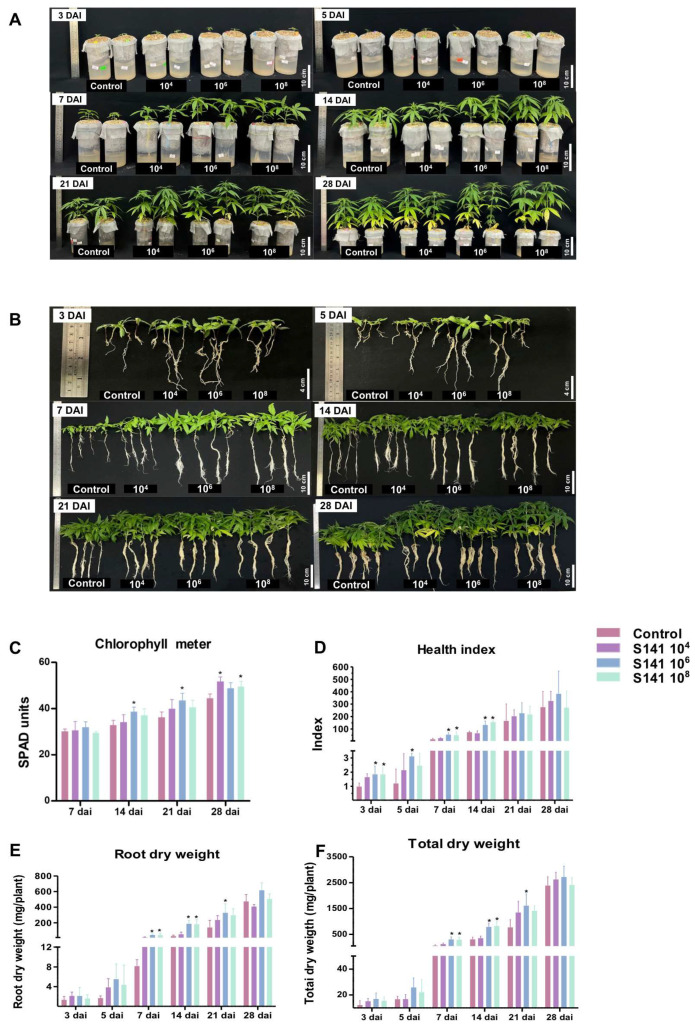
Evaluation of cannabis growth profiles after inoculation with *B. velezensis* S141 under laboratory conditions. Images were taken to inspect the growth of cannabis plants cultivated in Leonard’s jars after removal of the growth medium (**A**,**B**). Parameters representing the cannabis growth profiles including chlorophyll content (**C**), heath index (**D**), root dry weight (**E**), and total dry weight (**F**) were examined. Bars display means ± SD calculated from biological triplicate (n = 4), and asterisks denote statistically significant differences between treatment groups and control group (*p* < 0.05).

**Figure 3 plants-13-02971-f003:**
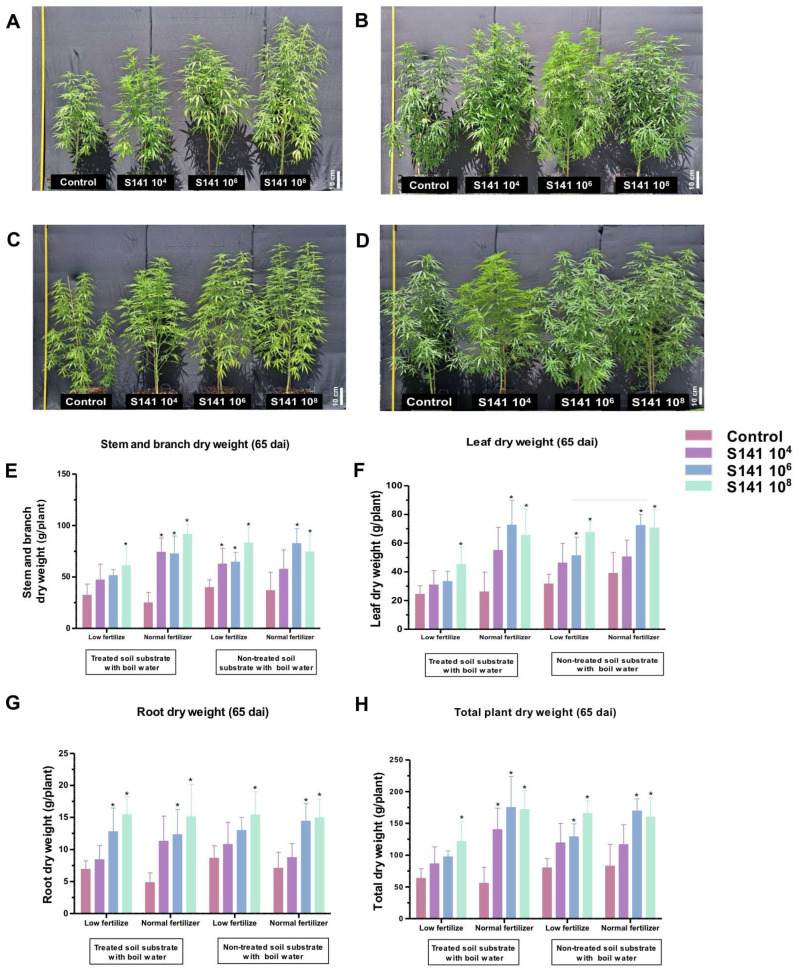
Assessment of cannabis growth patterns following inoculation with *B. velezensis* S141 in a greenhouse environment. The S141-inoculated and non-inoculated cannabis were cultivated in soil treated with boiled water containing either low fertilizer (**A**) or normal fertilizer (**B**) or untreated soil comprising low fertilizer (**C**) or normal fertilizer (**D**), from which features images of individually cultivated cannabis plants in pots were collected. Specific growth parameters, including leaf dry weight (**E**), stem and branch dry weight (**F**), root dry weight (**G**), and total dry weight (**H**), were monitored. Bars indicate mean values ± SD (n = 5) from three biological replications. Significant differences between control group are indicated by asterisks (*p* < 0.05).

**Figure 4 plants-13-02971-f004:**
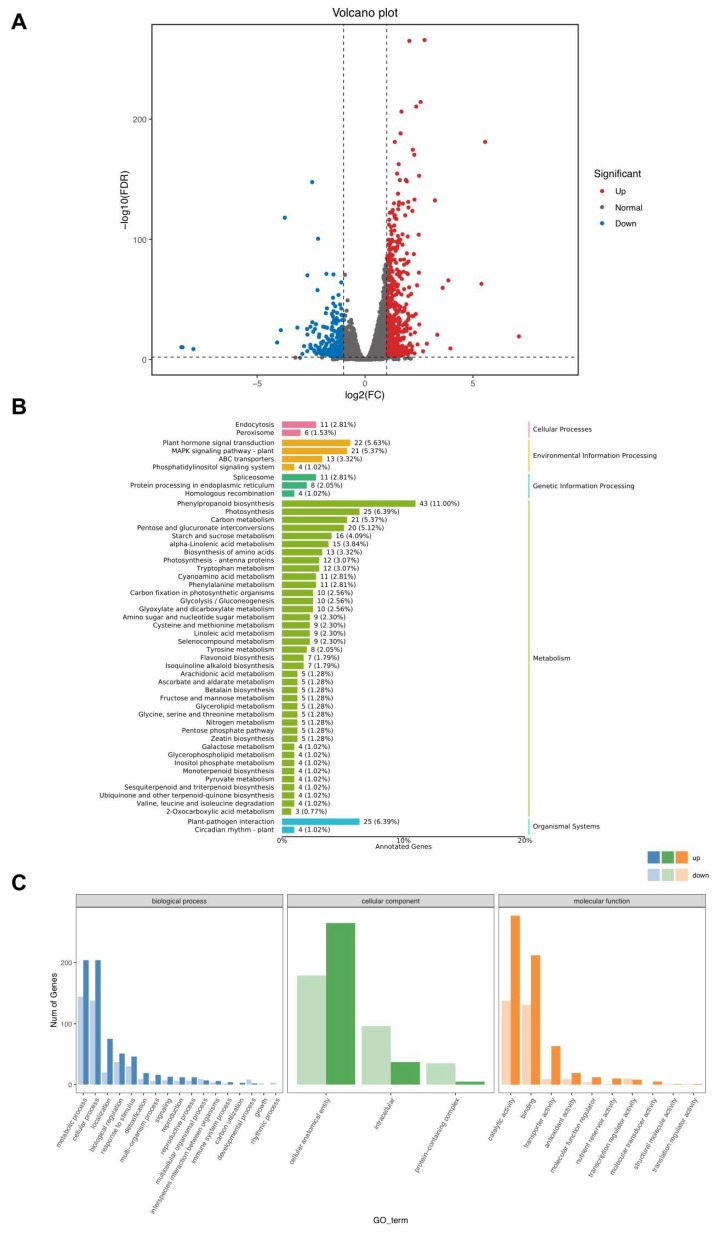
Transcriptomic analysis of S141-inoculated and non-inoculated cannabis. (**A**) A volcano plot displays differentially expressed genes, in which red- and blue-colored dots indicate up- and downregulated genes, whereas grey-colored spots display nondifferentially expressed genes, demonstrating a log2(Fold change) of more or less than 1 between libraries generated from inoculated and non-inoculated cannabis with an adjusted *p*-value < 0.05. (**B**) KEGG pathways classify these genes into cellular processes (pink), environmental information processing (orange), genetic information processing (dark green), metabolism (green), and organismal systems (blue). (**C**) Gene Ontology organizes them into biological processes (blue), cellular components (green), and molecular functions (orange), wherein dark and light colors represent categories with upregulated and downregulated genes, respectively.

**Figure 5 plants-13-02971-f005:**
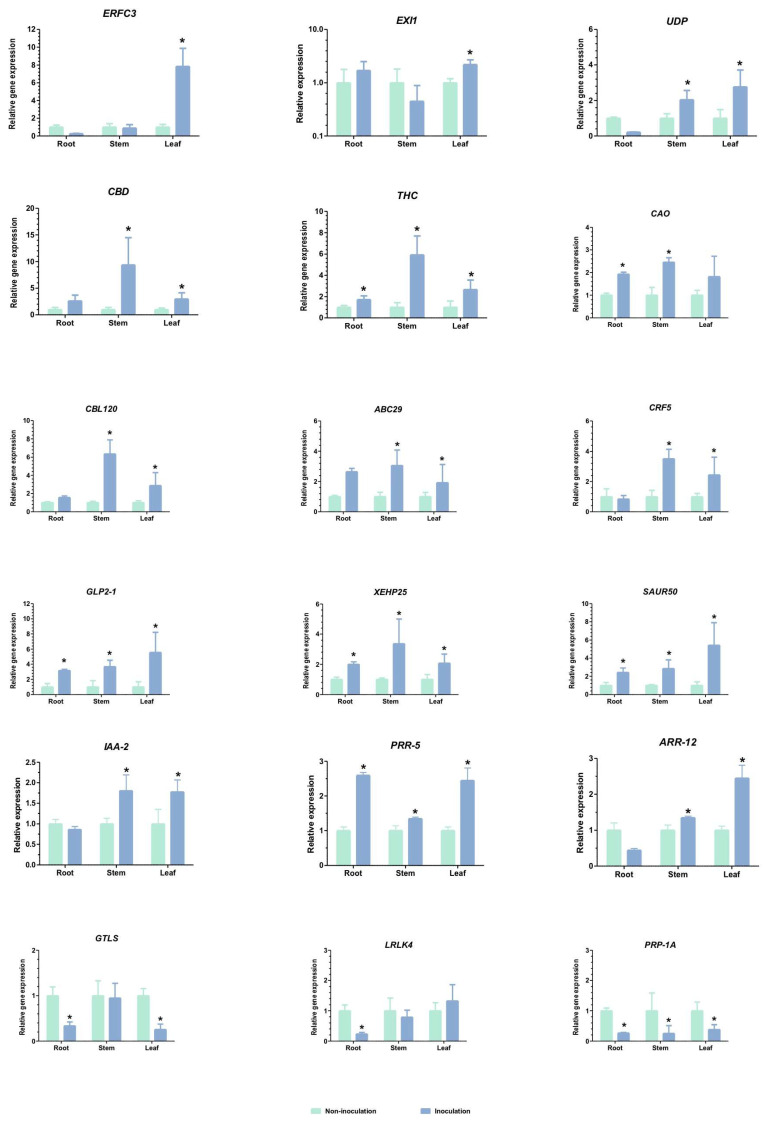
The qRT-PCR analysis of differentially expressed genes obtained from RNA-seq. Eighteen genes related to plant-growth-promoting impacts in cannabis were chosen, where the relative expression was calculated using the 2^−ΔΔCt^ method and normalized against actin, an internal control [19,20]. Bars indicate means ± SD analyzed from three biological replications (n = 4), and asterisks indicate significant differences at *p* < 0.05.

**Figure 6 plants-13-02971-f006:**
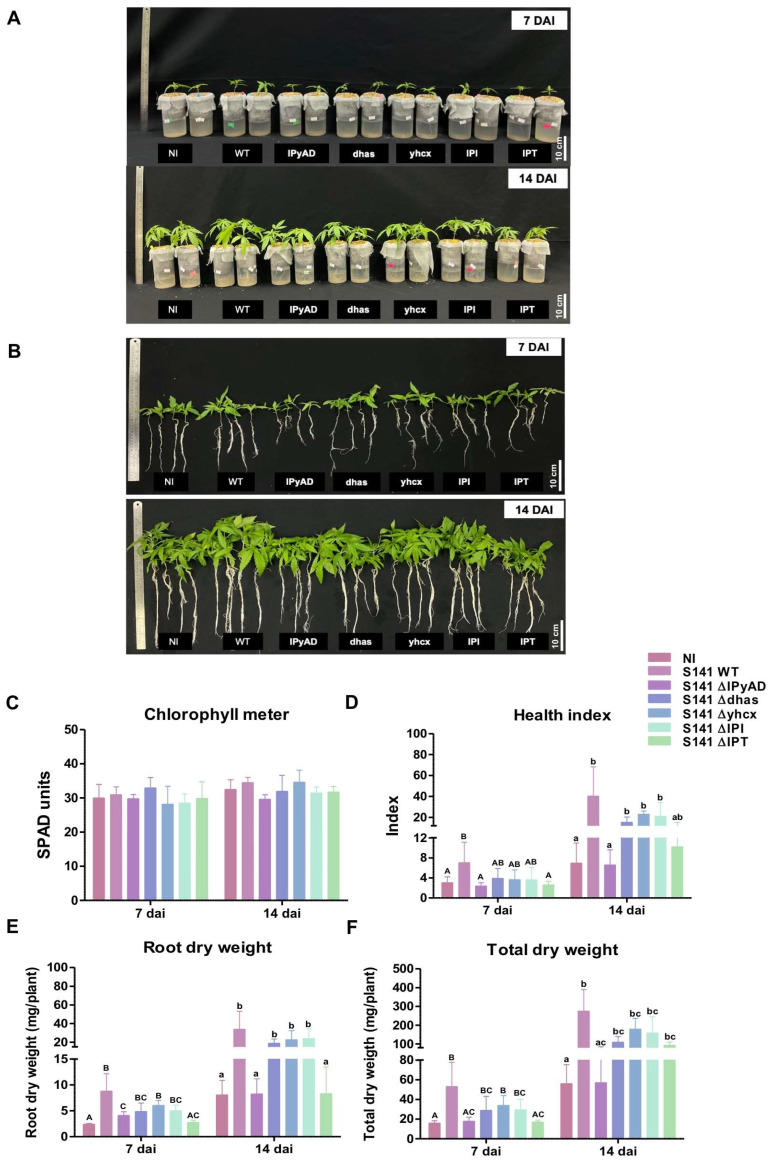
Examination of cannabis growth profiles following inoculation with *B. velezensis* S141 mutants. Visual inspection of the cannabis plants cultured in Leonard’s jars was performed by taking images of the plants after removing the growth medium (**A**,**B**). Parameters representing the cannabis growth profiles including chlorophyll content (**C**), heath index (**D**), root dry weight (**E**), and total dry weight (**F**) were examined. Bars display means ± SD calculated from biological triplicates (n = 4), and different letters indicate significant differences between treatment groups (*p* < 0.05).

**Figure 7 plants-13-02971-f007:**
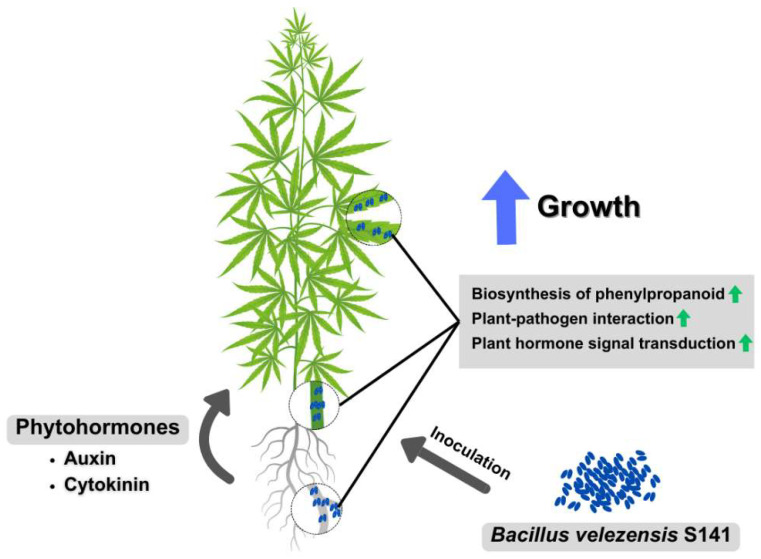
Schematic overview of mechanisms of cannabis growth promotion by *Bacillus velezensis* S141. S141, an endophytic cannabis bacterium, promotes cannabis growth by producing phytohormones and triggering genes involved in the biosynthesis of phenylpropanoid, plant–pathogen interaction, and plant hormone signal transduction pathways. This figure was created using https://www.canva.com/ (accessed on 21 August 2024).

## Data Availability

Data available upon request.

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
