# Peer review of "Mechanisms of Cannabis Growth Promotion by Bacillus velezensis S141"

_plants, 2024, doi:10.3390/plants13212971_

Round 1
Reviewer 1 Report
Comments and Suggestions for Authors
The article (research) is interesting and all its sections are congruent among them but it presents some important weaknesses:
i) The Methods ought to be improved. Authors never mentioned the number of seeds/seedlings/replicates used per treatments/controls, which is basic for the credibility of the results obtained.
ii) The statistical section has to be improved, detailing the tests used for comparing what parameter vs another parameter.
iii) Figures´ titles and legends have to be improved. The reader ought to completely understand them without reading the article or a part of it.
iv) References sections must to be checked according to the Authors Guide.
v) For this kind of research, the traditional way of presentation is better for achieving a better understanding of it: Introduction, Objectives, Materials and Methods, Results, Discussion, Conclusions and References.
Please, see file attached.

Author Response
Reviewer #1: The article (research) is interesting and all its sections are congruent among them but it presents some important weaknesses:
- The Methods ought to be improved. Authors never mentioned the number of seeds/seedlings/replicates used per treatments/controls, which is basic for the credibility of the results obtained.
Answer: Thank you for your comments. The number of seeds/seedlings/replicates used per treatments/controls has been explained in Materials and Methods. (Lines 357-376)
- The statistical section has to be improved, detailing the tests used for comparing what parameter vs another parameter.
Answer: Thank you for your comment. The statistical section has been re-written. Moreover, the detailing of the tests used for comparing has been explained in figure legends. (Lines 467-468, 664-666, 678-680, 687-688, 722-723, and 728-729)
- Figures´ titles and legends have to be improved. The reader ought to completely understand them without reading the article or a part of it.
Answer: Thank you for your suggestion. Figures´ titles and legends have been improved. (Lines 663, 675, 682-683, 691, 720, and 725)
- References sections must to be checked according to the Authors Guide.
Answer: Thank you for your comment. They have been modified in accordance with your recommendation.
- For this kind of research, the traditional way of presentation is better for achieving a better understanding of it: Introduction, Objectives, Materials and Methods, Results, Discussion, Conclusions and References.
Answer: Thank you for your suggestion. We have arranged the manuscript in the order recommended by the Plants journals and previous research articles published in Plants including Introduction, Results, Discussion, Conclusions, Materials and Methods, and References.

Reviewer 2 Report
Comments and Suggestions for Authors
The title of the manuscript is 'Transcriptomic profiling of Cannabis sativa supplemented with the plant-growth-promoting bacteria: Bacillus velezensis S141', however, there is only one Figure in the text about the results of transcriptome sequencing. Too many Figures are used to illustrate the beneficial effects of B. velezensis S141 and its mutant forms on cannabis growth in the laboratory and in the greenhouse, which does not seem to correspond exactly to what the title refers to.
Other Points:
1) In Figures 2 and 3, significance analyses should analyze the significance between each treatment, not just compared to the control. Also, in Figure 3, 'S141 108' seems to grow the best, but it doesn't in line with the statistics, why?
2) Figure 4's pixels are so low that I can't see any information clearly.
3) In Figure 5, When the authors performed transcriptome sequencing, only the roots were used for testing, so why did they choose stem and leaf to perform qPCR validation? After all, the authors were randomly selecting genes for validation, so adding stem and leaf data seems unnecessary.
4) Line 100 and Line 105, Full names of ‘DAI’ and ‘HI’.
5) Line 198, ‘indole-3-acetic acid (IAA)’ should be ‘IAA’.
6) Line 259, gene name should be italic.
7) Line 351, Line 385…, ‘mL’ or ‘ml’? Please select one and keep it consistent throughout the manuscript.
Author Response
Reviewer #2: The title of the manuscript is 'Transcriptomic profiling of Cannabis sativa supplemented with the plant-growth-promoting bacteria: Bacillus velezensis S141', however, there is only one Figure in the text about the results of transcriptome sequencing. Too many Figures are used to illustrate the beneficial effects of B. velezensis S141 and its mutant forms on cannabis growth in the laboratory and in the greenhouse, which does not seem to correspond exactly to what the title refers to.
Answer: Thank you for your comment. The title of the manuscript have been improved as your suggestion. (Lines 2-3)
- In Figures 2 and 3, significance analyses should analyze the significance between each treatment, not just compared to the control.
Answer: Thank you for your suggestion. According to the analysis of graphs in Figures 2 and 3, no significant different was observed when compared between S141 inoculation groups. So, the asterisks has been added only in the significant different between control and S141 inoculation groups.
Also, in Figure 3, 'S141 108' seems to grow the best, but it doesn't in line with the statistics, why?
Answer: Sorry for our mistake. S141 inoculation at 106 and108 CFU/ml has been mentioned in the text. (Lines 121-126)
- Figure 4's pixels are so low that I can't see any information clearly.
Answer: Sorry for our mistakes. They have been corrected accordingly to your suggestion.
- In Figure 5, When the authors performed transcriptome sequencing, only the roots were used for testing, so why did they choose stem and leaf to perform qPCR validation? After all, the authors were randomly selecting genes for validation, so adding stem and leaf data seems unnecessary.
Answer: Thank you for your comment. We collected RNA samples during the first 7 days after inoculation of the bacterial culture. We initially expected that the bacteria would mainly be present in the roots. However, as depicted in Figure 1 of the experimental results, bacteria were found in all cannabis tissues. This discovery led us to investigate whether the genes responsible for promoting cannabis growth would be expressed in other cannabis tissues, stem and leaf. So, the expression of genes were analyzed in all cannabis tissues.
- Line 100 and Line 105, Full names of ‘DAI’ and ‘HI’.
Answer: Thank you for your comment. They have been modified as suggestion. (Lines 99 and 101)
- Line 198, ‘indole-3-acetic acid (IAA)’ should be ‘IAA’.
Answer: Thank you for your comment. They have been modified in accordance with your recommendation. (Lines 62)
- Line 259, gene name should be italic.
Answer: Thank you for your comment. They have been modified though out the manuscript.

Reviewer 3 Report
Comments and Suggestions for Authors
In this manuscript (plants-3123197) entitled "Transcriptomic profiling of Cannabis sativa supplemented with the plant-growth-promoting bacteria: Bacillus velezensis S141" submitted to Plants, Phirom Aunkam and colleagues have evaluated the effects of PGPB Bacillus velezensis S141 on Cannabis sativa growth and utilized transcriptomic analysis to identify responsive pathways. Authors’ findings emphasize the potential of S141 as a sustainable growth promoter for cannabis and provide insight into the underlying pathways it influences. This research is interesting and convincing, but this present manuscript needs revisions before publication.
Major points:
1. For Figure 6, Authors indicated that “Significant differences between treatment groups are denoted by asterisks (P < 0.05)” in the legend. But I cannot find asterisk in the figure.
2. Authors should consider to include a Figure 7 to depict the model about the effects of PGPB Bacillus velezensis S141 on Cannabis sativa in the revision.
Minor points:
1. Authors need to standardize references according to the Plants template. For instance, the Abbreviation instead of full name of “Frontiers in Plant Science” (Reference 2) should be employed in the revision.
2. Full names of abbreviations like qPCR and DAI should be spelt out at their first appearance. Authors should check all abbreviations employed in the manuscript.
Author Response
Reviewer #3: In this manuscript (plants-3123197) entitled "Transcriptomic profiling of Cannabis sativa supplemented with the plant-growth-promoting bacteria: Bacillus velezensis S141" submitted to Plants, Phirom Aunkam and colleagues have evaluated the effects of PGPB Bacillus velezensis S141 on Cannabis sativa growth and utilized transcriptomic analysis to identify responsive pathways. Authors’ findings emphasize the potential of S141 as a sustainable growth promoter for cannabis and provide insight into the underlying pathways it influences. This research is interesting and convincing, but this present manuscript needs revisions before publication.
Answer: Thank you for your comment.
Major points:
- For Figure 6, Authors indicated that “Significant differences between treatment groups are denoted by asterisks (P < 0.05)” in the legend. But I cannot find asterisk in the figure.
Answer: Sorry for our mistakes. We have made the necessary changes to align with your suggestion. Moreover, the asterisk has been changed to be a different letters for comparing each treatment. (Lines 729)
- Authors should consider to include a Figure 7 to depict the model about the effects of PGPB Bacillus velezensis S141 on Cannabis sativa in the revision.
Answer: Thank you for your comment. Figure 7 has been constructed as suggestion.
Minor points:
- Authors need to standardize references according to the Plants template. For instance, the Abbreviation instead of full name of “Frontiers in Plant Science” (Reference 2) should be employed in the revision.
Answer: Sorry for our mistakes. We have made changes based on your suggestions.
- Full names of abbreviations like qPCR and DAI should be spelt out at their first appearance. Authors should check all abbreviations employed in the manuscript.
Answer: Thank you for your comment. They have been modified in accordance with your recommendation. (Lines 24 and 101)

Reviewer 4 Report
Comments and Suggestions for Authors
This paper demonstrates that Bacillus velezensis S141, known as a plant growth-promoting bacterium (PGPB), also has growth-promoting effects on Cannabis sativa. Furthermore, the mechanisms by which it promotes growth are explored through RNA-seq and inoculation of S141 mutants deficient in plant hormone synthesis genes. Therefore, this manuscript appears to contain several novel insights. However, there are some crucial pieces of information that are unclear, which need to be revised to complete the review. Below are my comments:
Comment1:
Fig. 6 appears to show inoculation experiments with S141 mutants deficient in plant hormone synthesis genes, but the labels for figures EFGH are incorrect, showing "control, S141 104…". Please correct these labels and accurately describe the results and discussion sections.
Comment 2:
What genes are targeted in the qPCR shown in Fig. 1? Additionally, the sequences of the primers used should be provided.
Comment 3:
The title of this paper is " Transcriptomic profiling of Cannabis sativa supplemented with the plant-growth-promoting bacteria: Bacillus velezensis S141" but it seems to mainly reflect the results in Fig. 5. Therefore, I recommend changing the title to reflect the overall content of the paper or to indicate the main findings revealed by the transcriptome analysis and inoculation experiments with S141 and its mutants.
Comment 4:
All the plant photographs include a ruler, but the size cannot be determined from them. A scale bar should be created based on the ruler.
Comment 5:
Regarding the greenhouse cultivation experiments (Fig. 3), what is the reason for the heat treatment of the soil and the changes in nutrient status?
Comment 6:
It would be beneficial to include information in the introduction about which plant and which tissue S141 was isolated from.
Author Response
Reviewer #4: This paper demonstrates that Bacillus velezensis S141, known as a plant growth-promoting bacterium (PGPB), also has growth-promoting effects on Cannabis sativa. Furthermore, the mechanisms by which it promotes growth are explored through RNA-seq and inoculation of S141 mutants deficient in plant hormone synthesis genes. Therefore, this manuscript appears to contain several novel insights. However, there are some crucial pieces of information that are unclear, which need to be revised to complete the review. Below are my comment.
- Fig. 6 appears to show inoculation experiments with S141 mutants deficient in plant hormone synthesis genes, but the labels for figures EFGH are incorrect, showing "control, S141 104…". Please correct these labels and accurately describe the results and discussion sections.
Answer: Sorry for our mistakes. We have made the necessary changes to align with your instructions.
2.What genes are targeted in the qPCR shown in Fig. 1? Additionally, the sequences of the primers used should be provided.
Answer: Thank you for your comment. The primer used in in the qPCR shown in Fig. 1 was designed from non-coding sequence from Bacillus velezensis S141 genome sequence (AP018402). The sequences of the primers used has been provided in Table S4.
3.The title of this paper is " Transcriptomic profiling of Cannabis sativa supplemented with the plant-growth-promoting bacteria: Bacillus velezensis S141" but it seems to mainly reflect the results in Fig. 5. Therefore, I recommend changing the title to reflect the overall content of the paper or to indicate the main findings revealed by the transcriptome analysis and inoculation experiments with S141 and its mutants.
Answer: Thank you for your comment. The title of the manuscript have been improved as your suggestion. (Lines 2-3)
- All the plant photographs include a ruler, but the size cannot be determined from them. A scale bar should be created based on the ruler.
Answer: Thank you for your comment. A scale bar has been added as your suggestion.
- Regarding the greenhouse cultivation experiments (Fig. 3), what is the reason for the heat treatment of the soil and the changes in nutrient status?
Answer: Thank you for your comment. We wanted to investigate whether reducing the abundance of other bacteria through the use of 100°C boiling water and reducing fertilizer could more clearly demonstrate the potential of inoculating S141 bacteria to promote cannabis growth. The results aligned with our expectations.
- It would be beneficial to include information in the introduction about which plant and which tissue S141 was isolated from.
Answer: Thank you for your comment. The S141 inoculation has been added in the introduction. (Lines 74-76)

Round 2
Reviewer 1 Report
Comments and Suggestions for Authors
Authors have done all the corrections suggested.
Though, there are some scientific names that must include the Authority and the family. It is important to check the GBIF DATA BASE for updating the scientific names recorded until now.
Please, see file attached.

Reviewer 2 Report
Comments and Suggestions for Authors
The authors have answered all of my concerns.
